# Grape Berry Detection and Size Measurement Based on Edge Image Processing and Geometric Morphology

**Lufeng Luo** [1]**, Wentao Liu** [1]**, Qinghua Lu** [1,*]**, Jinhai Wang** [1]**, Weichang Wen** [1]**, De Yan** [1] **and Yunchao Tang** [2,3]

[1] School of Mechatronics Engineering and Automation, Foshan University, Foshan 528000, China; luolufeng@fosu.edu.cn (L.L.); liuwentao96@163.com (W.L.); cswjh@fosu.edu.cn (J.W.); weichangwen806@163.com (W.W.); 2111951002@stu.fosu.edu.cn (D.Y.)

[2] College of Urban and Rural Construction, Zhongkai University of Agriculture and Engineering, Guangzhou 510225, China; ryan.twain@zhku.edu.cn

[3] Foshan-Zhongke Innovation Research Institute of Intelligent Agriculture and Robotics, Foshan 528000, China

\* Correspondence: qhlu@fosu.edu.cn; Tel.: +86-186-6639-2056

**Abstract:** Counting grape berries and measuring their size can provide accurate data for robot picking behavior decision-making, yield estimation, and quality evaluation. When grapes are picked, there is a strong uncertainty in the external environment and the shape of the grapes. Counting grape berries and measuring berry size are challenging tasks. Computer vision has made a huge breakthrough in this field. Although the detection method of grape berries based on 3D point cloud information relies on scanning equipment to estimate the number and yield of grape berries, the detection method is difficult to generalize. Grape berry detection based on 2D images is an effective method to solve this problem. However, it is difficult for traditional algorithms to accurately measure the berry size and other parameters, and there is still the problem of the low robustness of berry counting. In response to the above problems, we propose a grape berry detection method based on edge image processing and geometric morphology. The edge contour search and the corner detection algorithm are introduced to detect the concave point position of the berry edge contour extracted by the Canny algorithm to obtain the best contour segment. To correctly obtain the edge contour information of each berry and reduce the error grouping of contour segments, this paper proposes an algorithm for combining contour segments based on clustering search strategy and rotation direction determination, which realizes the correct reorganization of the segmented contour segments, to achieve an accurate calculation of the number of berries and an accurate measurement of their size. The experimental results prove that our proposed method has an average accuracy of 87.76% for the detection of the concave points of the edge contours of different types of grapes, which can achieve a good edge contour segmentation. The average accuracy of the detection of the number of grapes berries in this paper is 91.42%, which is 4.75% higher than that of the Hough transform. The average error between the measured berry size and the actual berry size is 2.30 mm, and the maximum error is 5.62 mm, which is within a reasonable range. The results prove that the method proposed in this paper is robust enough to detect different types of grape berries.

**Keywords:** image processing; particle detection; pit detection; fast radial symmetric transformation; cluster search; least square method

## 1. Introduction

The production scale and consumption of grapes are increasing year by year. Large-scale mechanized production is gradually replacing labor [1,2]. Calculating the number of berries in each bunch of grapes is a key component of estimating the yield of grapes, and the ratio of the diameter to the volume of each bunch of grape berries is also one of the many factors in evaluating the quality of grapes. Traditional detection is time-consuming and can easily cause estimation errors. To improve its detection efficiency and detection quality, researchers have been seeking solutions using computer vision and image processing [3–5].

In recent years, there are mainly two methods based on 3D information and on 2D images in the use of machine vision to estimate the number of grape berries [6,7]. Huerta [8] and Rist [9] used 3D point cloud equipment to scan the 3D information of grape spikes to obtain parameters such as the geometry and structure of the grapes to reconstruct the grape phenotype to estimate the number of grape berries [10]. Although the detection accuracy has been improved, they rely on the 3D scanning equipment in the laboratory to avoid problems in complex environments. Kicherer used berry analysis tools to obtain 2D images of the number, size, and volume of grape berries [11], but they are destructive and depend on the perforated metal plate under laboratory conditions. Liu reconstructed 3D berries based on the sparsity factor on the 2D color image of red grapes, with an average accuracy of 87.6% [12], but they can only process conical or cylindrical grapes, and the radius of the berries needs to be manually defined using their method. Liu also designed software for estimating the number of berries for a single image of black grapes and Niagara grapes, but the robustness of the estimation of the number of grape berries is not high [13]. Aquino proposed an algorithm based on mathematical morphology and pixel classification for berry characteristics [14,15]. Although these methods can estimate grape berries and grape yield, the number of grape berries detected largely deviates from the true value.

With the rapid development of deep learning, a large-scale global map of orchard picking can be obtained to adapt to the dynamic and complex orchard environment, and also makes it possible to detect the number of grapes under complex conditions [16]. Śkrabánek proposed the DeepGrapes detection model [17] to improve the recognition effect of grape berries at low resolution. Coviello designed a GBCNet network [18] with VGG16 as the backbone for the detection of grape yield, but its detection effect was average. In instance segmentation, Zabawa improved the recognition accuracy of grape berries by introducing additional "edge" categories as its feature information [19,20]. Nellithimaru proposed a new grape yield estimation method that combines instance segmentation and SLAM to obtain grape information [21], which improves the accuracy. However, the above mentioned deep neural network-based methods still have the problem of low robustness when analyzing characterization information such as the size of grape berries from dense and unconstrained grape images [22].

The size of the grape berries can provide important information for assessing the stage of maturity. First, the background of a single bunch of grapes must be segmented in a natural environment [23,24] to accurately identify the grape berries and accurately detect the number and size of the berries and other characterization information [25,26]. Researchers have conducted a plethora of studies on this. Xiao proposed a method based on brightness extreme value and edge information to detect red grapes, but it can only detect fruit particles with the same radius and fixed [27]. Liu Z proposed a method to detect the size of fruit particles based on gradient features [28], which has a low recognition rate for berries inside grapes. Zhou uses Hough transform to detect red grape berries [29], but Hough transform has a lot of false detections. Both Cubero and Behera use edge information to detect their size and other parameters, and their detection process is destructive to grape berries [30,31]. For the precise detection of objects with disorderly stacked structures, Langlard proposed a method based on pattern recognition, but its detection effect is very poor when applied to disorderly stacked grape berries [32].

In the grape berry count and berry size measurement, the problems are that it cannot be applied in a complex environment, the estimation accuracy of the number of berries is not high, and the size of a single berry cannot be accurately measured [33]. We propose an algorithm that can be used to detect various types of rounds or round-like grape berries. The algorithm first uses edge detection algorithms [34] and image preprocessing methods to obtain grayscale images and binary images of a single bunch of Niagara grapes, black grapes, and red grapes. According to the gradient value of the gray image, the interest points on the grape berries are extracted [35,36], and then the berry edge sequence (berry edge pixels) in the binary image is searched, and the berry edge is segmented into contour segments by the concave point detection algorithm [37]. The continuous iterative

calculation is performed on each section of contour to obtain the centroid point of the contour [38], and then the berry edge contour section is correctly combined [39], and finally, the grape berry is fitted by least squares and its diameter is detected.

The remainder of this paper includes four sections. Section 2 introduces how to collect data sets and image preprocessing methods. Section 3 introduces the edge contour segmentation algorithm of berries. Section 4 introduces the grouping of contour segments of grape berries. Section 5 introduces the experimental results and discusses them. Finally, the conclusion is described in Section 6. The detection algorithm framework of grape berries is shown in Figure 1.

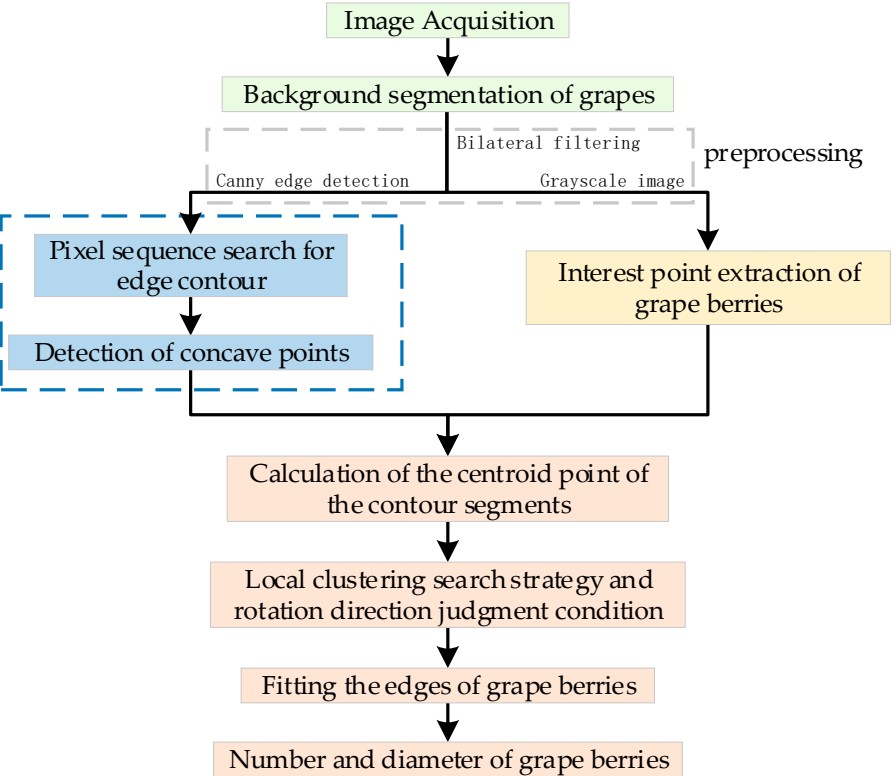

**Figure 1.** Detection algorithm framework of grape berries.

The contributions of this paper can be summarized as follows:

(1) This work proposes a detection model for multiple types of rounds or round-like grape berry counting and berry size. It solves the problem that the characterization information of grape berries cannot be accurately detected.

(2) Before the berry edge contour segment grouping strategy, the corner detection algorithm and the fast radial symmetry change algorithm are introduced. These can help realize the segmentation of the overlapping edges of berries.

(3) This paper proposes an algorithm for combining contour segments based on clustering search strategy and rotation direction determination, which realizes the correct reorganization of segmented contour segments. Finally, according to the obtained edge information of a single berry, the berry is fitted, and its size is estimated.

## 2. Datasets and Preprocessing

The grape images used in this experiment were collected from Guangdong Luonan Ecological Park. The images of Niagara grapes, black grapes, and red grapes were taken in clear weather conditions. According to the experimental requirements, this paper designed an image acquisition system including a camera and tripod. The maximum resolution of the camera is 1280 × 480; examples are shown in Figure 2. During image acquisition, Niagara

grapes, black grapes, and red grapes with different plumpness were randomly photographed at different angles and plant distances; examples are shown in Figure 3. Each time the grape images are collected, they are named after the date of collection. A hundred images of each type of grape are taken, resulting in a total of 300 images. The algorithm proposed in this study corresponds to the written custom code, and then the Hough transform uses the code library, and they are all written in MATLAB (version 2020b).

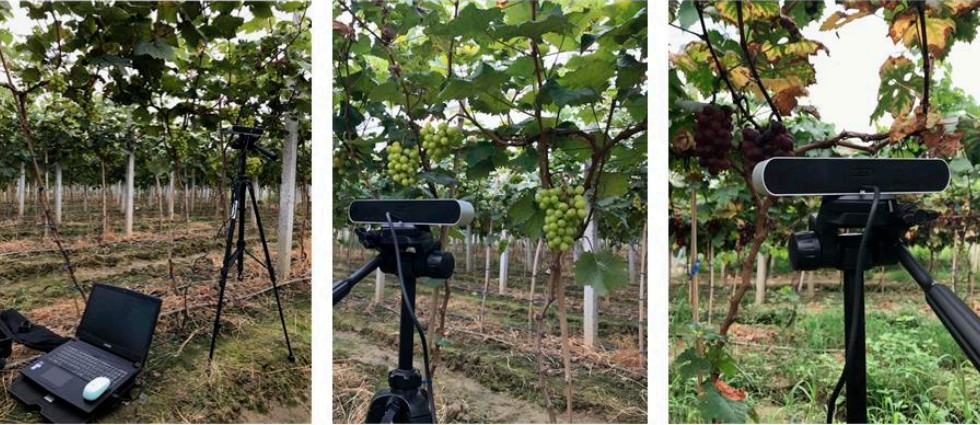

**Figure 2.** Shooting environment and equipment image.

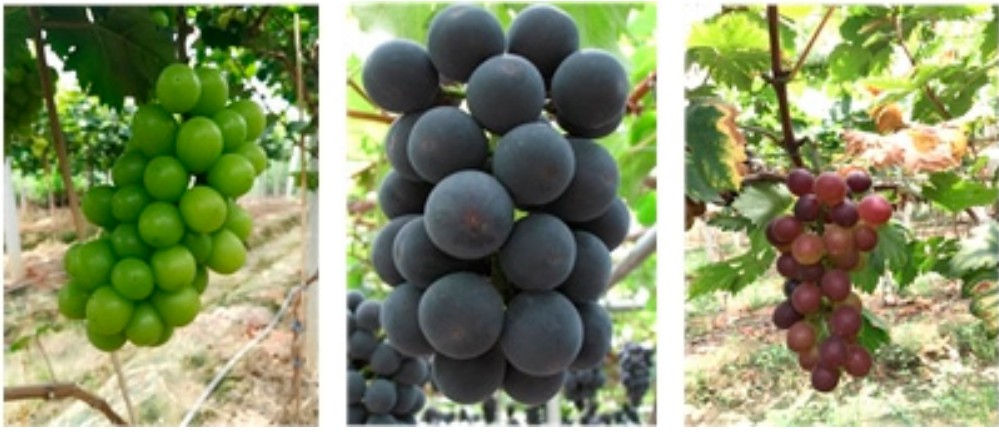

**Figure 3.** The image of grape strings of various types.

A single grape is obtained after removing the background through the perfect background removal algorithm. To reduce interference effects such as noise on the grape berries, this paper uses bilateral filtering to remove the interference in the image while retaining the edge information. The filtered RGB image is converted into a gray image; examples are shown in Figure 4b. According to the user-defined threshold and standard deviation, Canny edge detection model is applied to the gray image to extract the edge information of grape berries. The final effect is shown in Figure 4c.

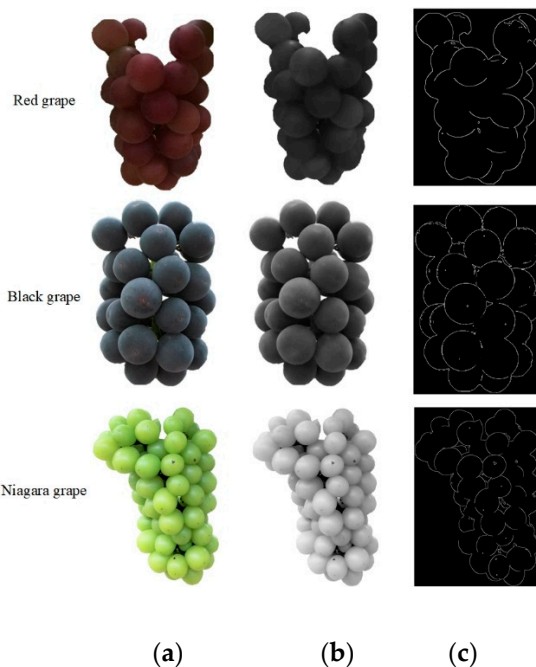

**(a)**　　　　**(b)**　　**(c)**

**Figure 4.** Image preprocessing. (**a**) stands for original image; (**b**) stands for gray image; (**c**) stands for edge information map of Canny detection.

## 3. Edge Contour Segmentation Algorithm of Berries

In this section, we describe the method of extracting points of interest on grape berries and the method of segmenting overlapping edge contours based on the concave points on the berries.

### 3.1. Interest Point Extraction

In the gray image, the potential points of interest in the grape berries are found through Fast Radial Symmetry Transformation (FRST) [26,27]. The radius range of the radial symmetry transformation is set to n. The pixel point p is compared with each surrounding pixel point. If the point where the distance is n and the gradient direction $g(p)$ is the same as the positive influence point $p_{+ve}(p)$, the resulting expression is as follows:

$$p_{+ve}(p) = p + r\left(\frac{g(p)}{\|g(p)\|}n\right) \tag{1}$$

If the opposite gradient direction is the negative influence point $p_{-ve}(p)$, the resulting expression is as follows:

$$p_{-ve}(p) = p - r\left(\frac{g(p)}{\|g(p)\|}n\right) \tag{2}$$

The $r$ in the formula rounds each vector element to the nearest integer.

We then judge each pixel point. If it is a positive influence point, we add 1 at the corresponding position of direction projection $O_n$ and $\|g(p)\|$ at the corresponding position of amplitude projection $M_n$. If it is a negative influence point, we minus 1 at the corresponding position of direction projection and $\|g(p)\|$ at the corresponding position of amplitude projection. The formula is as follows:

$$O_n(p_{+ve}(p)) = O_n(p_{+ve}(p)) + 1 \tag{3}$$

$$O_n(p_{-ve}(p)) = O_n(p_{-ve}(p)) - 1 \tag{4}$$

$$M_n(p_{+ve}(p)) = M_n(p_{+ve}(p)) + \|g(p)\| \tag{5}$$

$$M_n(p_{-ve}(p)) = M_n(p_{-ve}(p)) - \|g(p)\| \tag{6}$$

We calculate $O_n$ and $M_n$ and add a two-dimensional Gaussian filter $A_n$ to obtain $S_n$, which makes the points of interest more obvious. Combining $S_n$ forms a radial filter responder $S$ to obtain potential points of interest for the berries. The formula is as follows:

$$S_n = A_n \frac{M_n(p)}{k_n} \left( \frac{\left| \tilde{O}_n(p) \right|}{k_n} \right)^2, \tilde{O}_n(p) = \begin{cases} O_n(p) & O_n(p) < k_n \\ k_n \end{cases}, k_n = \begin{cases} 8 & n = 1 \\ 9.9 \end{cases} \tag{7}$$

$$S = \max_{n \in N} S_n \tag{8}$$

### 3.2. Segmentation of Overlapping Edges of Berries

This paper uses the edge contour search algorithm to find the edge information of the berry, under the binary image obtained by the Canny edge detection. Then, the concave points between overlapping berries are identified by the corner recognition algorithm based on adaptive dynamic curvature, and then the berry edge contour is divided into contour segments according to the concave point position.

#### 3.2.1. Pixel Sequence Search for Edge Contour of Grape Berry

A 3 × 3 mask is created with the position of the first edge pixel on the leftmost side of the binary image as the center. Then, the edge pixels of the berries are searched for on the binary image according to the vertical search method. The template is moved, and the next boundary pixel P obtained from the search is placed in the center of the mask. An example of pixel sequence search for the edge contour of a grape berry is shown in Figure 5.

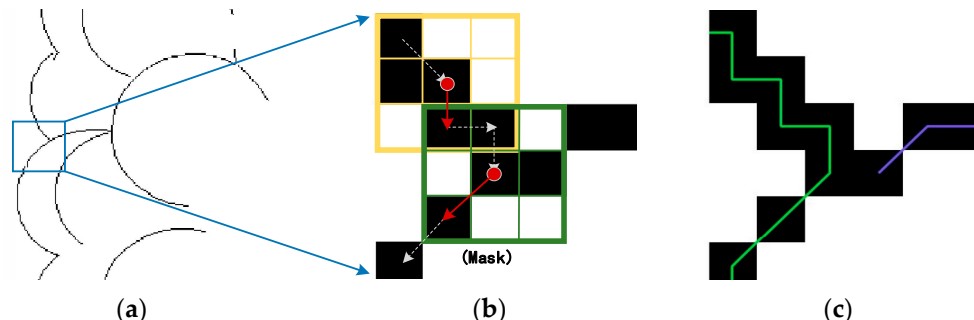

(**a**)　　　　　　　　　(**b**)　　　　　　　　　(**c**)

**Figure 5.** An example of edge contour segmentation algorithm of berries. (**a**) stands for binary graph; (**b**) stands for local image, the red dot is the center of the mask, and the red and gray arrows indicate the search direction; (**c**) Search results with different colors representing edge contours.

The search step is repeated to until all berry edge pixels are searched. Since the searched berry edge sequence contains the noise of small connected regions, they are subsequently removed.

#### 3.2.2. Detection of Concave Points between Overlapping Berries

The grape berries overlap each other, and the concave point of the overlapping part of the outline corresponds to the intersection between the grape berries. For the pixel sequence extracted from the edge contour of the grape berry in the previous step, $C = \{C_1, C_2, \cdots, C_h\}$, the curvature scale space corner detection algorithm with adaptive threshold and dynamic support area is used. Firstly, at a low scale, the contour point $C$ is convolved with the Gaussian function $g(c, \sigma)$, and then the curvature $K$ of the contour is calculated. The point with the maximum local curvature on the contour is taken as the candidate point. The definition of curvature $K$ is as follows:

$$X(c, \sigma) = x(c) \cdot g(c, \sigma), Y(c, \sigma) = y(c) \cdot g(c, \sigma) \tag{9}$$

$$K(c,\sigma) = \frac{\dot{X}(c,\sigma)\ddot{Y}(c,\sigma) - \ddot{X}(c,\sigma)\dot{Y}(c,\sigma)}{\left(\dot{X}(c,\sigma)^2 - \dot{Y}(c,\sigma)^2\right)^{3/2}} \tag{10}$$

where $x, y$ represents the coordinate position of the contour point. $\sigma$ represents the scale deviation. $\dot{X}$, $\dot{Y}$ is the first partial derivative with respect to $\sigma$. $\ddot{X}, \ddot{Y}$ is the second partial derivative with respect to $\sigma$.

Then, the mean curvature of candidate points in the dynamic support area are calculated, and then the adaptive threshold is set according to the mean curvature. Finally, the best concave point is selected according to the threshold and angle range. The definition is as follows:

$$\overline{K} = \frac{1}{L_1 + L_2 + 1}\sum_{c-L_2}^{c+L_l} K \tag{11}$$

$$T_K(u) = 1.5 \times \overline{K} \tag{12}$$

where $u$ is the candidate point, $\overline{k}$ is the mean value of the curvature of $u$ in the dynamic support area, and $L_1$ and $L_2$ are the size of the dynamic support area.

We then divide the contour of the berry edge into a contour segment with the concave point as the boundary. The grape contour segment should resemble a circular arc with smooth curvature value changes, but the interference of noise causes some edge curvature values to change strangely. Therefore, we compare the curvature values of each point of each contour with the curvature values of the previous pixel and the next pixel. If the difference is greater than a certain threshold, it is a singular point. According to our many experimental tests, the best threshold is set to 0.015. The segmentation effect of berry edge contours with different degrees of overlap is shown in Figure 6.

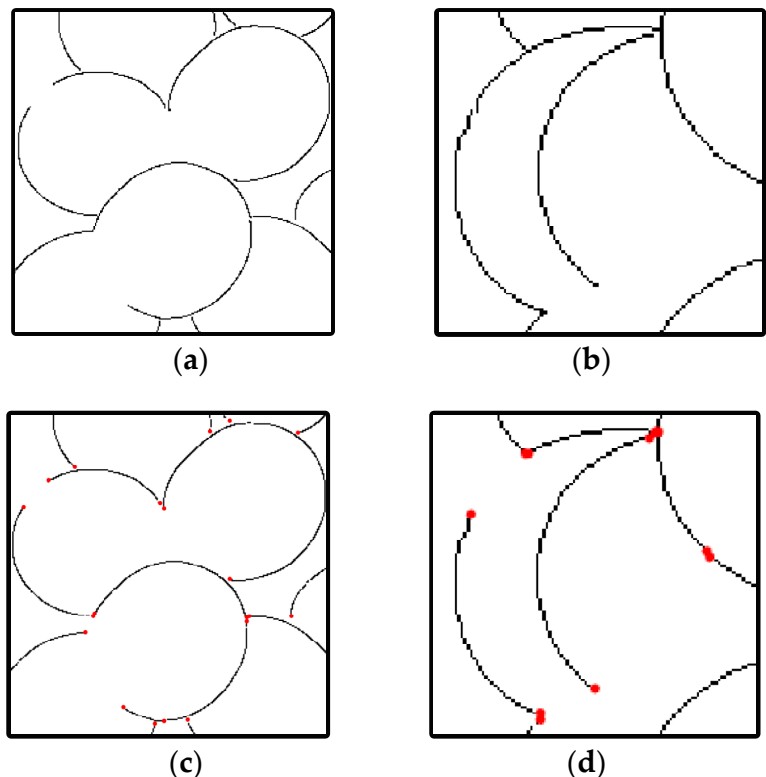

**Figure 6.** Concave point detection and edge contour segmentation of grape berries with different overlapping degrees. (**a**,**b**) are the berry edge contours with different degrees of overlap. The red points on (**c**,**d**) indicate the division points of the grape contour.

## 4. Correct Grouping of Contour Segments of Grape Berries

In this section, the calculation method of the centroid point of the contour segment is based on the random sampling consensus algorithm [29]. The local clustering search strategy and the rotation direction judgment condition are described to combine the contour segments of the same berry.

### 4.1. Calculation of the Centroid Point of the Contour Segments

Due to the irregular overlap between grape berries and incomplete edge information of some berries, the same grape berry detected in the previous section may be divided into multi-segment contours. In this paper, an algorithm based on random sampling consensus is used to calculate the centroid points and the direction of the contour of each segment.

The contour segment of the edge of the grape berry is like a circular arc. $S = \{s_1, s_2, \cdots, s_N\}$ is N sets of samples of contour segment. In a circular arc, three points that are not adjacent are selected arbitrarily to determine whether the three points are collinear, and if they are collinear, three points are randomly selected again. Assuming that $A$, $B$ and $C$ are three non-collinear points on an arc, a triangle is constructed from these three points. The intersection of the perpendicular bisector of the line segment $AB$ and $BC$ is the center of the circumcircle of the triangle, the center point is $O(x,y)$, and the radius is $r$. The vector $\overrightarrow{DO}$ of any point $D$ on the contour $S_i$ is directed to the centroid of the grape berry; an example is shown in Figure 7a. According to the random sampling consensus algorithm, it is estimated that the highest quality center point of the contour segment contains the most pixels and the closest point of interest; an example is shown in Figure 7b.

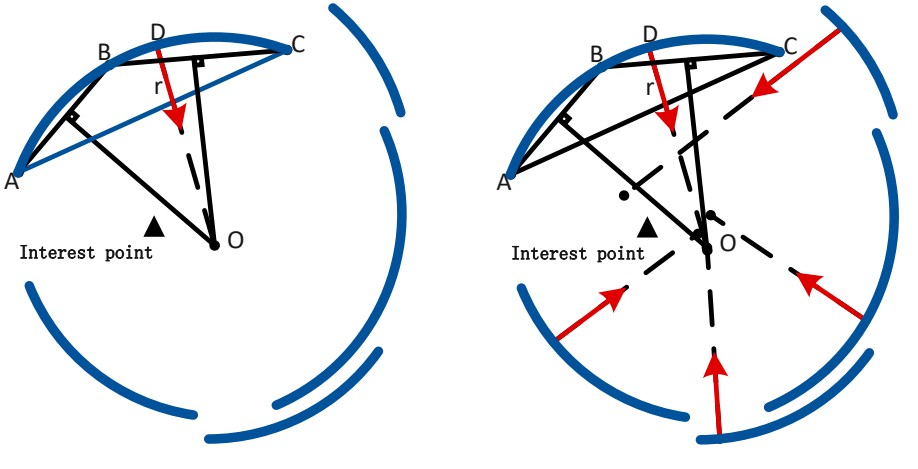

(**a**). Construct a triangle          (**b**). The center of each segment

**Figure 7.** The centroid point of the contour segment is calculated. The direction vector is represented by the red arrow.

### 4.2. Local Clustering Search Strategy and Rotation Direction Judgment Condition

To eliminate the potential false classification contour segments, this paper uses the local clustering search and rotation direction judgment method to combine all the contour segments belonging to the same grape, and then fits each group of contour segments by the least square method. Finally, according to the fitting results, the number of grape berries is calculated, and the diameter of the berries is measured.

This paper defines the clustering search allocation strategy of the contour segment samples. We calculate the Euclidean distance between the centroid point of each contour segment $O = \{O_1, O_2, \cdots, O_N\}$ and each point of interest, search all centroid points in the local range of the point of interest, and then mark the centroid point closest to the point of interest and the contour segment corresponding to the centroid point. Fuzzy search is performed in the local range of the marked centroid point. We search for the centroid point

that satisfies the radius $r$ of the contour segment within $r_j - R_1 \leq r \leq r_j + R_1$ and is closest to the marked centroid point and mark this centroid point and its contour segment. If there is no point of interest in a certain berry, we directly perform a fuzzy search. The formula is as follows:

$$d_{ij} = \min \|e_i - 0_j\|_2^2 < T_d \cup r_j + R_1 < r < r_j + R_2 \tag{13}$$

where $e_i$ represents the point of interest, $O_j$ represents the center point, $T_d$ represents the range of the local search, r should be the distance from the marked point to the marked contour segment, and $R_1$ is the optimal adaptive range that satisfies $r$.

We continue to iterate with this search strategy method to obtain a set of contour segments $\omega_i$, $i \in M$, and then calculate the center of mass $\mu_i$ of the group $\omega_i$, the formula is as follows. After searching, all contour segments are grouped into $\omega = \{\omega_1, \omega_2, \cdots, \omega_M\}$, $M < N$.

$$\mu_i = \frac{1}{|m|} \sum_{O \in \omega_i} O \tag{14}$$

where $m$ is the number of contour segments in the group.

Assuming that the starting point of any section of contour in $\omega_i$ is $a$ and the endpoint is $b$, the rotation angle and direction of the two-dimensional vector $l_1(\mu, a)$ to the two-dimensional vector $l_2(\mu, b)$ can be obtained. The formula is as follows.

$$div(l_1, l_2) = \frac{l_1(\mu, a) l_2(\mu, b)}{\|l_1(\mu, a)\| \|l_2(\mu, b)\|} \tag{15}$$

$$dir(l_1, l_2) = l_1(\mu, a) \times l_2(\mu, b) \tag{16}$$

The direction of the vector $\vec{l_1}$ to the vector $\vec{l_2}$ is used to determine whether the rotation direction of the contour is clockwise or counterclockwise. We know that the rotation direction of any continuous point of the closed curve is the same. We define the rotation direction of the contour $s_i$ in $\omega_i$ to be positive. Then, we search for the nearest contour $s_{i+1}$ around the endpoint $b$ of the contour segment $s_i$ in the $\omega_i$ group. The point close to the endpoint $b$ is the starting point of the contour line segment $s_{i+1}$. After that, we calculate the rotation angle and rotation direction that vector $l_2(\mu, b)$ to the start point of $s_{i+1}$, and determine the rotation direction. If it is positive, it is the same grape berry contour. If it is negative, it satisfies that the rotation angle that $l_2(\mu, b)$ to the start point of $s_{i+1}$ is less than 8°, the rotation direction that $l_2(\mu, b)$ to the endpoint of $s_{i+1}$ is positive, and the rotation angle that $l_2(\mu, b)$ to the endpoint of $s_{i+1}$ is greater than 50°, it is the same grape berry contour. Otherwise, it is not the same berry contour. Under this search strategy, the contour segments of the same grape berry are grouped together; examples are shown in Figure 8.

Finally, each group is a partial outline of a grape berry. Because the edge information of grape berries is not sufficient, we need to infer the outline of the occluded or missing part of the grape through the existing contour segment. The classic least-squares fitting method can fit the object by observing part of the information of the object. We calculate the number of grape berries based on the fitting results and then detect their complete size and other characterization information.

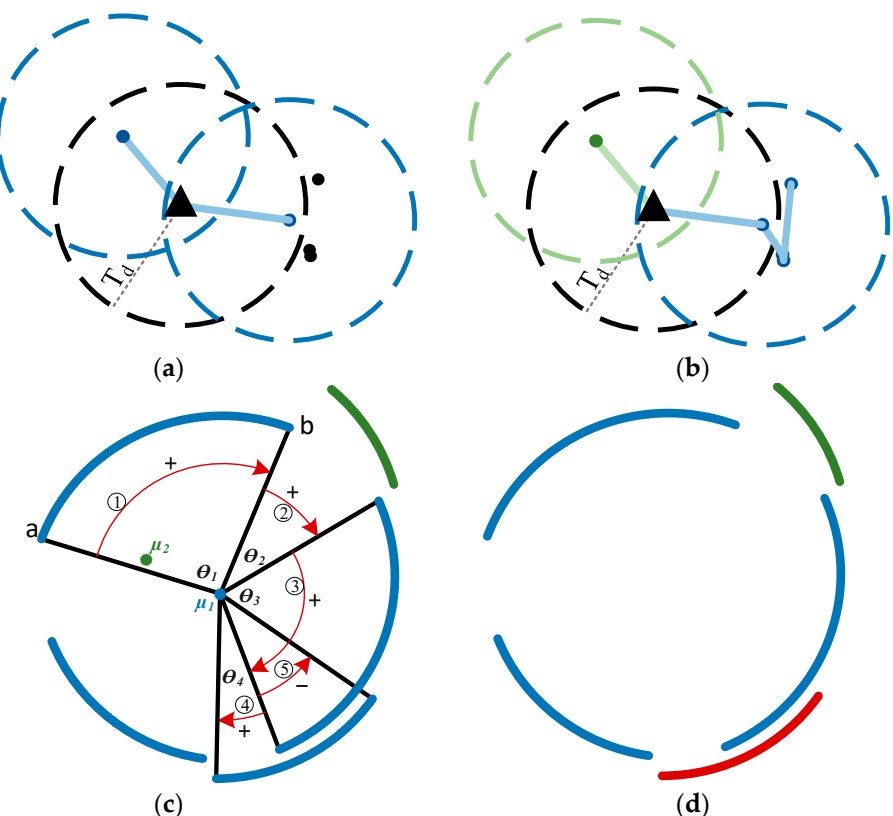

**Figure 8.** Schematic diagram of the local clustering search strategy and rotation direction judgment condition. (**a**) is to find the closest point from the point of interest, (**b**) is divided into two categories according to the search range and radius, (**c**) is the judgment of the false classification segment according to the rotation direction and rotation angle, (**d**) is the final contour grouping. Different groups are a different color, the dotted line is the local search range, the red arrow indicates the direction of rotation, and ($\theta_1$, $\theta_2$, $\theta_3$, $\theta_4$) is the angle of rotation.

## 5. Experiment Results and Discussion

### 5.1. Performance Measurement Evaluation Index

To verify that this method can correctly detect grape berries in the overlapping state, this paper uses a total of 300 pictures of three different types of grapes for evaluation experiments. We use the Positive Predictive Value (*PPV*), True Positive Rate (*TPR*), and the Average Distance of pixel (*AD*) from the detection point to the real point as the index to evaluate the performance of grape and berry contour segmentation. The Average Jaccard Similarity Coefficient (*AJSC*) is used as an indicator to evaluate the detection effect of berries on grape spikes. The formula is as follows:

$$PPV = \frac{TP}{TP + FP} \tag{17}$$

$$TPR = \frac{TP}{TP + FN} \tag{18}$$

$$JSC = \frac{E_s \cap E_t}{E_s \cup E_t} \tag{19}$$

$$AJSC = \frac{1}{\varepsilon} \sum_{i=1}^{\varepsilon} JSC \tag{20}$$

where *TP* is the true-positive rate, that is, the number of correct detections. *FP* is the false-positive rate, that is, the number of errors in detection. *FN* is the false-negative rate, that is, the number that is undetected. It is determined that the distance from the calculated

concave point to the actual concave point is within the predefined threshold $\rho_1$, then it is determined that it is a correctly detected concave point (*TP*), and the threshold is set to 2 pixels. $E_S$ is the predicted value, which is the detected grape berry; $E_t$ is the true value, which is the real grape berry. In this paper, the threshold of the *JSC* overlap ratio is set to 0.5. $\varepsilon$ is the number of samples, that is, the number of grape berries.

The size of the detected berry is the pixel size. The system is calibrated through the calibration board, the conversion coefficient is determined as $\Gamma$, and the pixel size is converted into the actual size in the space, the formula is as follows. It is compared with the manual measurement of the real size of the grape berry to verify the accuracy of the measurement of the size of the grape berry.

$$\Gamma = \frac{p_s}{p_d} \tag{21}$$

$$P_S = \Gamma \cdot P_D \tag{22}$$

where $p_s$ is the actual size of the calibration tool. $p_d$ is the pixel size of the calibration tool. $\Gamma$ is the conversion coefficient after calibration. $P_S$ is the actual size of the grape berry. $P_D$ is the pixel size of the grape berry.

### 5.2. Analysis of the Concave Ppoint Detection of Different Types of Grapes

Table 1 verifies the feasibility and practicability of the method in this paper to segment different types of grape contour segments. On 300 grape cluster samples, we analyze the performance of the corner detection algorithm with adaptive threshold and dynamic support for regional curvature through *PPV*, *TPR*, and *AD*.

**Table 1.** Detection and analysis of pits of different types of grape berries.

| Grape Type | TPR/% | PPV/% | AD/Pixel |
|---|---|---|---|
| Red grape | 90.47 | 82.97 | 3.47 |
| Black grape | 85.21 | 94.23 | 3.01 |
| Niagara grape | 87.61 | 92.92 | 3.34 |

It can be seen from the results that after Canny edge detection and grape berry edge contour sequence search, the *TPR* value of the concave point detected by the red grape is the highest, which is 90.47%, but its *PPV* value is lower, which is 82.97%. The main reason for this is that compared with other grapes, the red grape has a larger berry size and a larger gap between the berry and the berry, so the accuracy rate is high. For the red grape berries in the shadow area, the surface color is similar, and the texture information is weak, resulting in a relatively small gradient value of the contour edge, so relatively complete edge information cannot be detected. Therefore, the red grape has a false negative rate during the concave point detection.

The test results show that the *PPV* evaluation values of Niagara grapes and black grapes are similar, reaching 94.23% and 92.92%, respectively. However, for black grapes and Niagara grapes, the area covered by individual berries exceeds 94%, and the local curvature between individual berries is not the largest, so the concave point cannot be detected. As long as the *TPR* is lower than that of red grapes, black grape berries or Niagara grape berries overlap closely, causing the berries to be compressed and deformed. Although we removed most of the stains through filtering algorithms and small connected areas, and also used edge curvature thresholds to make the contour segments relatively smooth to reduce the impact of compression deformation, because the noise is dense, it cannot be completely filtered out. These conditions affect the judgment of the algorithm, and the noise is detected as false concave point. The *TPR* values of black grapes and Niagara grapes were 85.21% and 87.61%, respectively. In terms of the average pixel distance between the detection point and the real point, *AD* of all grapes is less than 3.5 pixel. The algorithm

can accurately detect and achieve high detection accuracy for the concave point areas of various types of grapes.

Through the analysis mentioned above, it can be concluded that the surface color and texture at the intersection of the two grapes, the noise of the grapes, and the tightness between the grapes all have an influence on the correct detection of the grape edge information. The result of the contour segment detection proposed in this paper is shown in Figure 9. For each noise filtering stage, a large amount of noise is eliminated, which increases the accuracy of identifying grape contours. This method can correctly detect each segment of grape contours in most case. It is also very robust to the detection effect with a high degree of overlap and obvious fruit-grain intersections, and the location of the detection points is reasonable.

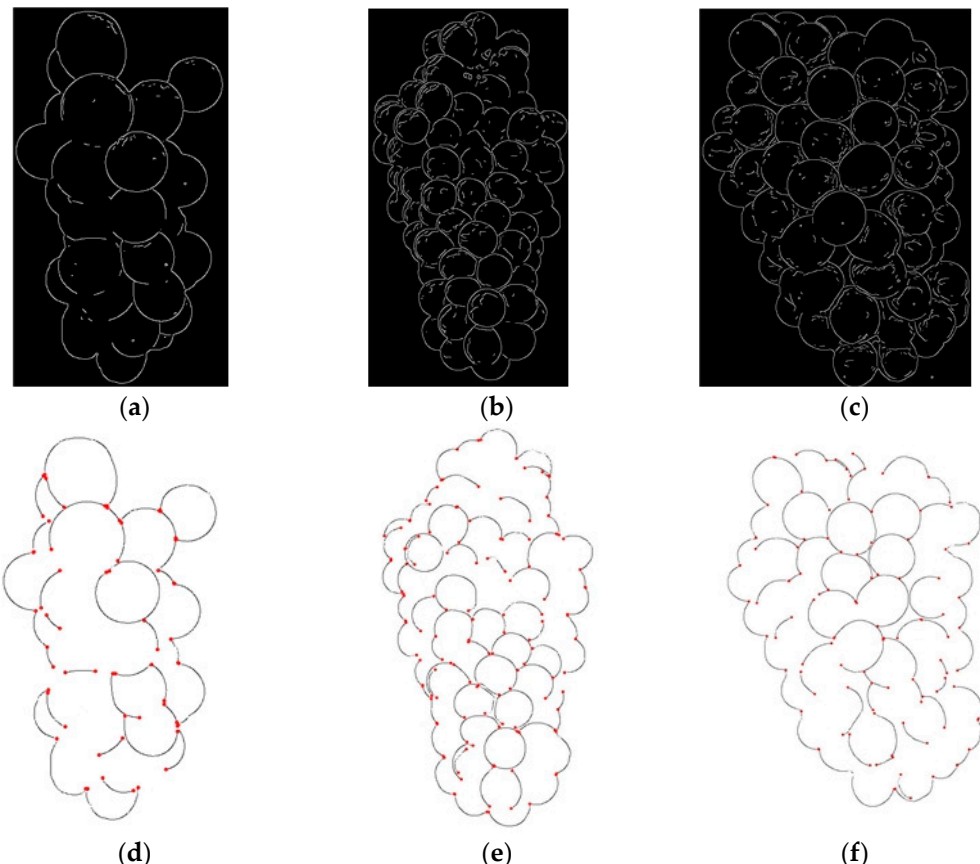

**Figure 9.** Different types of grape berry contour segment differentiation graph. (**a**) is a binary map of red grapes, (**d**) is a detection map of red grapes contour pits, (**b**) is a binary map of purple-black grapes, and (**e**) is a pit detection map of purple-black grapes. (**c**) is the binary map of green grapes, (**f**) is the detection map of the contour pits of the green grapes, and the red dots represent the final pits of the contour segment of the grapes.

Although a very small part of the noise edges is classified as an effective contour segment and the grape contour segments shorter than the correctly detected grape contour segments are removed due to the increase in accuracy, the detection result still matches the original image. It shows that in terms of the segment the contour segments of different types of grapes, the proposed algorithm in this paper has very high robustness and accuracy.

### 5.3. Analysis of the Results of Checking the Number of Berries on Grape

In order to further verify the actual berry number detection effect of the proposed algorithm under different types of grapes, in this experiment, three types of grapes with different numbers, different sizes, different shapes, and different overlapping densities

were selected and compared with Hough transform to detect grape berries. The edge threshold and sensitivity of the Hough transform were adjusted precisely for different types of grapes. The specific process and results of this algorithm are shown in Figure 10. The centroid points of the segmented grape berry contour segments were calculated, and different contour segments were marked with different colors; examples are shown in Figure 10f.

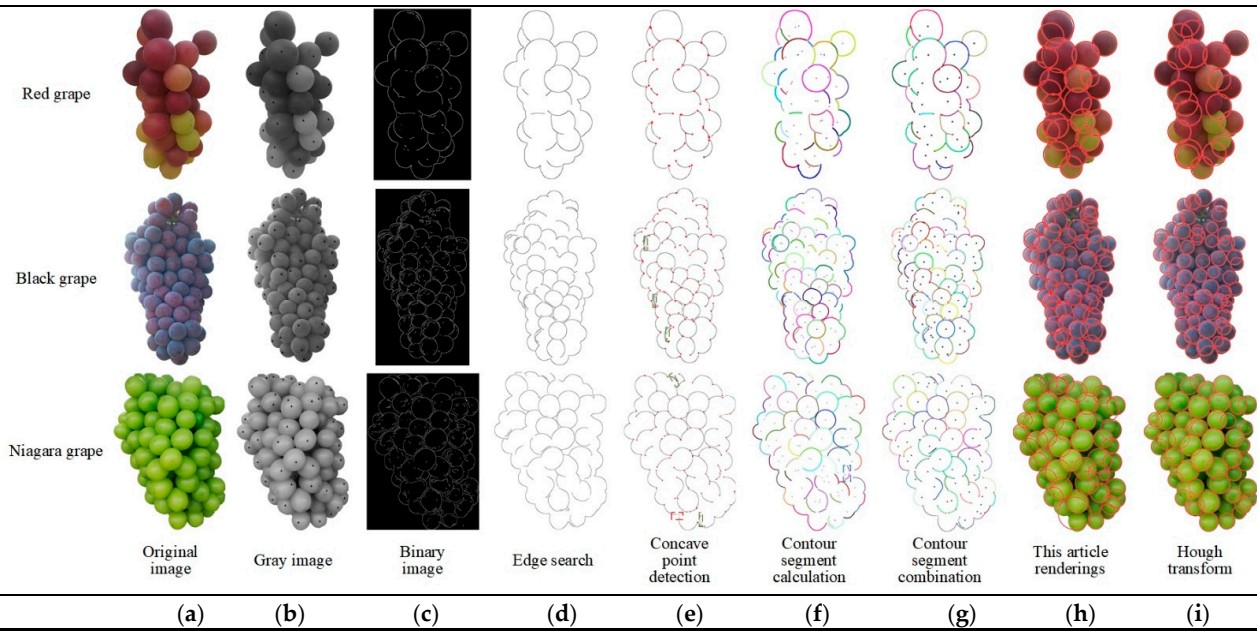

**Figure 10.** Model detection effect and comparison diagram under different types of grapes. (**b**), "·"represents the point of interest of the fruit; The green dashed frame and the red dashed frame of (**e**) respectively represent the noise and undetected pits that are mistaken for the edge of the grape, and the red dots indicate the position of the detected pits; The blue dashed box in (**f**) indicates the position of the edge arc segment with too little information, the blue dot indicates the position of the centroid point calculated by the contour segment, and the contour segments of different colors indicate different segmented contour segments; The contour segments of different colors in (**g**) represent the contours of different groups of fruit grains, and "·" represents the position of the center point calculated for each group of contour segments; The red circles in (**h**,**i**) respectively represent the particles detected by the method in this paper and the particles detected by the Hough transform.

Due to the excessive covering and compression between the grape berries, the berries are deformed, so different contour segments on the same berry will produce deviations and interference occurs between the contour segments of different berries. Under the same conditions, different $R_1$ values have different effects on the grouping performance of contour segments. The influence of r on the detection effect is shown in Figure 11.

The analysis shows that different $R_1$ has different effects on the TPR of grape berries. If the $R_1$ value is too small, the same berry contour segment will be divided into different groups, and the number of berries will be incorrectly detected, leading to the decrease of *TPR*. If the $R_1$ value is too large, the contour segments of non-berries will be divided into one group, resulting in inaccurate berry detection. $R_1$ value in the range of [25,35] can accurately divide the contour segment of the same grape berry into a group that reduce the error grouping, and the estimated centroid is also very close to the center of the grape berry. Therefore, the search strategy in this paper can limit the relevance of search objects and provide an accurate grouping of contour segments.

By mapping, the grape berries detected in this paper and the grape berries detected by the Hough transform model to the original Figure 10a and compared with the actual grape berries on the original image; the mapping results are shown in Figure 10h,i. The detection performance of the two models on different types of grape berry was compared and analyzed, and the results are presented in Table 2.

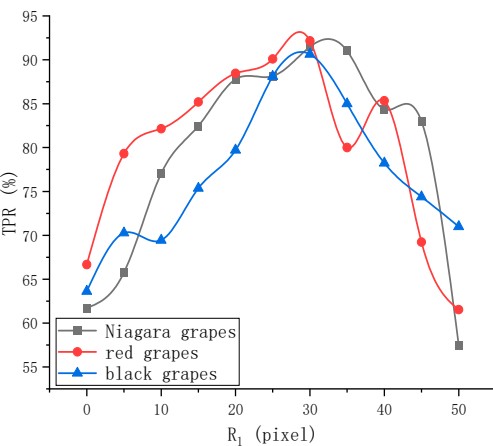

**Figure 11.** Effect of different $R_1$ on grape pulp detection. $R_1$ is the fuzzy search range near the marked centroid point.

**Table 2.** Comparison of detection results of different methods on various grape berries.

| Methods | Grape Type | TPR/% | PPV/% | AJSC/% | Time/s |
|---|---|---|---|---|---|
| our Algorithm | Red grape | 92.15 | 87.59 | 88.33 | 17.848 |
| | Black grape | 90.63 | 84.61 | 90.46 | 36.651 |
| | Niagara grape | 91.48 | 86.23 | 86.29 | 42.433 |
| Hough transform detection circle | Red grape | 84.71 | 76.46 | 86.91 | 7.389 |
| | Black grape | 87.49 | 83.08 | 88.17 | 5.796 |
| | Niagara grape | 87.80 | 81.79 | 86.36 | 7.067 |

It can be concluded from Table 2 that this method has the highest *TPR* and *PPV* scores for red grapes, which are 92.15% and 87.59%, respectively. Compared with Niagara grapes and black grapes, the noise of red grapes is less mistaken for contour segments, the area covered by other berries is smaller, and the number of berries per bunch is also less than that of other types of grapes. For red grapes, the *TPR* score of the precisely adjusted Hough transform model is 84.71%, and the *PPV* score is 76.46%. Compared with the Hough transform model, the model in this paper increases the *TPR* by 7.44% and the *PPV* by 11.13%. Through *AJSC*, it can be seen that the *AJSC* coefficient of the black grape berries reaches 90.46%, which is 2.29% higher than the Hough transform detection. The *AJSC* of black grapes is higher than that of Niagara grapes and red grapes because the shape of black grapes is closer to round than that of red grapes and Niagara grapes. The fitting result of the contour segment is closer to the contour of the real fruit. The recognition speed of red grapes is 17.848 s, and the recognition speed of black grapes and Niagara grapes is about 40 s. However, the recognition speed of this algorithm is still much slower than the Hough transform, which is 10.459 s, 30.855 s, and 35.366 s, respectively.

Through the analysis, it can be concluded that the reasons for the detection errors in the detection of the algorithm in this paper are the following. In the process of grouping grape berry contour segments, too large noise edges are mistaken as grape contour edges; examples are shown in the green dashed box in Figure 10e. Due to the stacking and squeezing of grapes berries, the contours show irregular arcs. During the calculation of the arcs, the center points of the arc segments of the same fruit are too far apart and the radius difference is too large, so they are not divided into a group in the end; examples are shown in the medium blue dotted box in Figure 10f. Moreover, for grape berries with the too-large, obscured area, information of the edge is too small, which leads to the inaccurate fitting of individual grape berries.

There are two main reasons why some grape berries are not detected. First, the information of the edge contour segment of the grape berry is too small and is removed

as noise. Second, the local maximum curvature of the fruit-grain intersection area is not obvious, and the concave point is not detected; examples are shown in the red dashed box in Figure 10e.

Because the algorithm in this paper performs a large number of calculations on each detected edge pixel, different types of grapes are different in size and quantity, resulting in different calculation speeds. Red grapes have fewer berries and fewer pixels, so the recognition speed is faster than that for black grapes and green grapes.

After the model is calculated, the contour segments of the same berry are marked with the same color, and different fruit pieces are distinguished by different colors; examples are shown in Figure 10g. The results of this experiment can show that for grapes of different types, different numbers of berries, different fruit sizes, different fruit shapes, and different overlapping densities, the number of grape berries can be accurately detected. As long as the edge contour can be detected and the contour segments can be accurately grouped, then the number of grape berries can be detected, as well as their location.

Compared with other methods, the grape berry detection method in this paper has high robustness to various types of irregularly overlapping and highly dense grape berries in most cases, but the detection speed is slower.

### 5.4. Discussion of Method Limitations

The method in this paper is to use computer vision to solve the round or quasi-circular grape berry counting and size measurement. For the detection of non-circular berries, the detection effect may not be the same as that of round or quasi-circular berries. Therefore, the elongated shape of grapes was tested. Taking typical ARRA grapes as an example, the robustness of the method is verified by detecting grape berries of different shapes. Figure 12 shows that for the elongated grape berries, the performance of our algorithm is weakened.

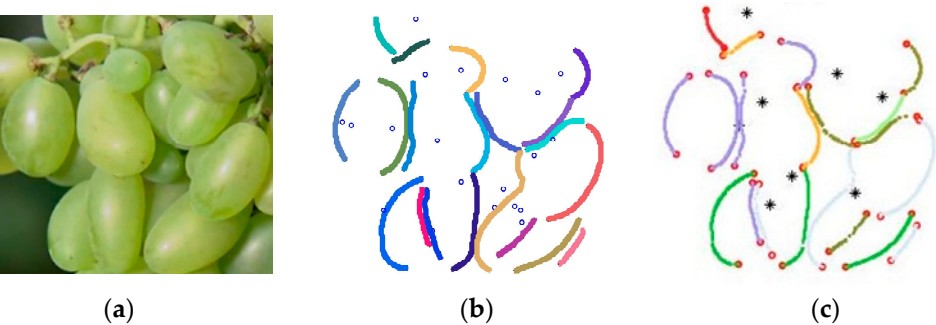

|      |      |      |
|:----:|:----:|:----:|
| (**a**) | (**b**) | (**c**) |

**Figure 12.** The model test results of ARRA grape berries. (**a**) is original image, (**b**) is contour segment calculation, (**c**) is contour segment combination. The red dot represents the concave point of the contour line segment of the grape berry; the blue dot represents the position of the centroid point of the contour segment; "✳" represents the position of the center point calculated for each group of contour line segments; different contour line segments are marked with different colors.

Firstly, the algorithm can detect the edge information of ARRA grapes and filter out most of the non-berry edge information such as stains and obtain relatively clear slender grape berries edges. However, for ARRA grapes, some areas with larger stains or darker shades cannot be eliminated by filtering algorithms and small connected area elimination algorithms. The edge contours of these areas will be classified as effective contours. Secondly, the overlapping part of the berry can accurately estimate the position of its concave point through the curvature algorithm, and then remove the singular pixels that are not smooth in the contour segment. However, because the edge of the berry is not in a circular arc shape and the local maximum curvature of the partially intersecting area is not obvious, it cannot accurately detect its concave point. Finally, the local clustering search strategy and the rotation direction judgment condition are to determine the berry group of the contour segment based on the arc information of the round berry. The edge

information of the slender grape berry presents an irregular shape. The algorithm in this paper cannot accurately combine the contour segments for the same berry.

In summary, the performance of the method in this paper has limitations on slender grape berries, and it is robust on rounds or round-like grape berries.

### 5.5. Berry Size Detection and Analysis on the Grape

Due to the camera, distance and other factors have an impact on the conversion of pixel size to actual size in space. This experiment uses the same camera, and the distance between the grape and the camera is constant. The conversion coefficient Γ is obtained through calibration, and then the system is verified by inverse inspection to determine that the error is less than 0.1 mm, and the system has good stability. We detected the actual berry size of each type of grape in space in a laboratory environment and compared the size of the berry detected in the randomly selected grape image with the size of the grape berry detected manually. The results are shown in Table 3.

**Table 3.** Comparison of the test results of grape berry size.

| Number | Manual Detection Diameter/mm | Image Detection Diameter/mm | Diameter Error/mm |
|---|---|---|---|
| 1 | 24.80 | 24.52 | 0.28 |
| 2 | 28.11 | 30.61 | 2.5 |
| 3 | 28.26 | 33.88 | 5.62 |
| 4 | 24.79 | 26.09 | 1.3 |
| 5 | 26.17 | 27.94 | 1.77 |
| 6 | 27.77 | 27.46 | 0.31 |
| 7 | 26.04 | 24.09 | 1.95 |
| 8 | 28.07 | 31.62 | 3.55 |
| 9 | 26.06 | 23.74 | 2.32 |
| 10 | 27.42 | 30.78 | 3.36 |
| Average | | | 2.30 |
| Max | | | 5.62 |

Through the analysis of Table 3, it can be concluded that the average error between the size of the grape berry by the method and the actual berry size measured is 2.30 mm in different samples of grapes, and the maximum error is 5.62 mm. According to statistical data, it can be found that the average error of the berry size detected by the method in this paper is relatively small, and the detection error of the berry size is relatively large when the berry outline information is less.

In general, the grape berry detection method in this article can measure the size of grape berries relatively accurately.

## 6. Conclusions

Based on the berry edge contour of a single grape after removing the background, this paper constructs a grape berry detection method based on concave point detection and optimal contour segment clustering. The specific conclusions are as follows. Bilateral filtering is used to remove interference noise, and Canny algorithm is used to extract grape edge contours. It is proposed to search through the pixel sequence of the edge contour of the grape berry, and then segment the berry contour segment by detecting the intersection between the overlapping grape berries. The average precision rate for different types of grapes is 87.76%, and the average recall rate is 90.04%; for red grapes The *TPR* of grapes is the highest, reaching 90.47%; the *PPV* of the black grapes is the highest, reaching 94.23%. It is proposed to combine the contour segments belonging to the same berry according to the points of interest and judgment conditions of the grape berries. When the value of R_1 is between [25,35], the contour segments of grape berries can be accurately combined. Compared with the Hough transform, the *TPR* of red grape berries

increased by 7.44%, and the *PPV* increased by 11.13%; the *AJSC* of purple grape berries increased by 2.29%. According to the obtained grape contour, the berry size is measured and converted into the actual berry size in the space. Compared with the real berry size, the average error is 2.30 mm, and the maximum error is 5.62 mm. In summary, the method proposed in this paper can detect characterization information such as the number and size of round or round-like grape berries in the natural environment. Although the noise, the degree of occlusion, and the curvature of the intersection will have a certain impact on the detection results, the detection accuracy of grape berries is improved. We can conclude that this method provides support for accurately estimating the weight, yield, and quality of individual grapes through visual perception technology in a natural environment.

**Author Contributions:** Conceptualization and methodology, L.L. and W.L.; writing, L.L. and W.L.; experiment, W.L., D.Y. and W.W.; review and editing, L.L. and Q.L.; supervision, J.W. and Y.T.; funding acquisition, L.L. All authors have read and agreed to the published version of the manuscript.

**Funding:** This work was supported in part by the National Natural Science Foundation of China under Grant 32171909, 51705365, the Guangdong Basic and Applied Basic Research Foundation under Grant 2020B1515120050, 2020B1515120070, the Guangdong key R & D Projects Grant 2020B0404030001 and the Scientific Research Projects of Universities in Guangdong Province under Grants 2019KTSCX197, 2018KZDXM074, and 2020KCXTD015.

**Institutional Review Board Statement:** Not applicable.

**Informed Consent Statement:** Not applicable.

**Data Availability Statement:** Not applicable.

**Acknowledgments:** We sincerely acknowledge the project funding support. We are also grateful for the efforts of all our colleagues.

**Conflicts of Interest:** The authors declare no conflict of interest.

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
