# Peer review of "Grape Berry Detection and Size Measurement Based on Edge Image Processing and Geometric Morphology"

_machines, doi:10.3390/machines9100233_

Round 1

Reviewer 1 Report

A new grape recognition algorithm is presented in this article. The algorithm works on the basis of data from a stereo camera. The algorithm is described in a scientific style and the results of practical experiments are presented. It is advisable to add information to the article in which programming language the algorithm is implemented. It is also advisable to add information about the speed of the algorithm. Can we use the algorithm in real-time?

Author Response

Dear Reviewers :

Thank you very much for your careful review and constructive suggestions with regard to my manuscript “Grape Berry Detection and Size Measurement Based on Edge Image Processing and Geometric Morphology (ID: ISSN 2075-1702)”. These comments are helpful for author to revise and improve the paper. I have studied comments carefully and tried my best to revise and improve the manuscript and made great changes in the manuscript according to the referees′ good comments. All the revised portions in the manuscript are used the “Track Changes” function of MS Word. The main corrections in the paper and the responds to the reviewer’s comments are as follows. I appreciate for Editors/Reviewers’ warm work earnestly, and hope that the corrections will meet with approval. Please feel free to contact me with any questions and I am looking forward to your consideration.

Dear reviewer 1:
1. Response to comment:
Main comments:

  • A new grape recognition algorithm is presented in this article. The algorithm works on the basis of data from a stereo camera. The algorithm is described in a scientific style and the results of practical experiments are presented. 
  • Response:Many thanks!
  • It is advisable to add information to the article in which programming language the algorithm is implemented.

Responds: In the first paragraph of section 2 of this article, information about the programming language of the algorithm implementation is added.

  • It is also advisable to add information about the speed of the algorithm. Can we use the algorithm in real-time?
  • Responds: In Table 2 of this article, information about the speed of this algorithm is added, and the speed of different algorithms is compared and analyzed in section 5.3. The recognition speed of red grapes is 17.848s, and the recognition speed of black grapes and Niagara grapes is about 40s. However, the recognition speed of this algorithm is still much slower than the Hough transform, which is 10.459s, 30.855s, and 35.366s respectively. It shows that the algorithm in this paper is to calculate each edge pixel detected, and different types of grapes are different in size and quantity, resulting in different calculation speeds. Compared with other algorithms, the speed of this algorithm is slower.

Methods

Grape type

TPR/%

PPV/%

AJSC/%

Time/s

our Algorithm

Red grape

92.15

87.59

88.33

17.848

Black grape

90.63

84.61

90.46

36.651

Niagara grape

91.48

86.23

86.29

42.433

Hough transform detection circle

Red grape

84.71

76.46

86.91

7.389

Black grape

87.49

83.08

88.17

5.796

Niagara grape

87.80

81.79

86.36

7.067

Reviewer 2 Report

The paper presents the detection and size estimation of grapes from images. The presented method is based on edge detection and geometric processing. The method is based on the geometric properties of circular arcs in the projected images. The method is compared with the Hough transform and seems to give good results. However, the use of circular arcs restricts the method to round grapes. Images of elongated or ellipsoid-shaped grapes will probably not result in as good results (e.g., images of the ARRA variety of grapes could be tested to evaluate this further).

I suggest that the author add a section about the limitations of the method.  The authors mention that there might be issues with "squeezed" grapes, but a section about the limitations, and ideas on how to possibly overcome these would be recommended to outline this further. Also, results from experiments with not exactly round grapes could be added.

Please also consider how your method would react to irregularities, such as stains on the grapes.

Please, also add complexity considerations of the algorithms.

As there is a variety of abbreviations used in the paper, I suggest adding a table of abbreviations and acronyms in the appendix.

Further minor comments (most of them are language-related):

  • line 78: remove "at home and abroad".
  • line 91: "grapes berries" -> "grape berries"
  • line 100: new paragraph after "detected."
  • line 110: incomplete sentences. Suggestion: "It solves the problem ..."
  • line 113: incomplete sentence.
  • line 118: consider using "estimated" instead of "measured"
  • line 122: --> "weather conditions"
  • line 124: "as shown in ..." -> "Examples are shown in ..."
  • line 133: add "The image of " (sentence is incomplete)
  • Figure 4: a, b, c are not visible in the figure.
  • line 144: In this section, we describe ... (add: we)
  • line 152: something is wrong with this sentence. further: "was" -> "is"
  • line 155: was --> is
  • line 160: something is wrong with this sentence: "influence point subtracts ..."
  • line 218: "to the test": I did not understand this.
  • line 370: "mention" -> "mentioned"
  • Figure 10: some of the symbols in the caption are not visible in the figure.
  • Caption of Figure 11: R_1

Author Response

Dear Reviewers :
Thank you very much for your careful review and constructive suggestions with regard to my manuscript “Grape Berry Detection and Size Measurement Based on Edge Image Processing and Geometric Morphology (ID: ISSN 2075-1702)”. These comments are helpful for author to revise and improve the paper. I have studied comments carefully and tried my best to revise and improve the manuscript and made great changes in the manuscript according to the referees′ good comments. All the revised portions in the manuscript are used the “Track Changes” function of MS Word. The main corrections in the paper and the responds to the reviewer’s comments are as follows. I appreciate for Editors/Reviewers’ warm work earnestly, and hope that the corrections will meet with approval. Please feel free to contact me with any questions and I am looking forward to your consideration.

Dear reviewer 2:
(1)The paper presents the detection and size estimation of grapes from images. The presented method is based on edge detection and geometric processing. The method is based on the geometric properties of circular arcs in the projected images. The method is compared with the Hough transform and seems to give good results. However, the use of circular arcs restricts the method to round grapes. Images of elongated or ellipsoid-shaped grapes will probably not result in as good results (e.g., images of the ARRA variety of grapes could be tested to evaluate this further).
Response:Many thanks!

(2)I suggest that the author add a section about the limitations of the method. results from experiments with not exactly round grapes could be added
Responds: This article adds a discussion about the limitations of the method in Section 5.4. A typical ARRA grape was used as an example to test the elongated shape of grapes. And made an analysis, the analysis results are as follows: First, the algorithm can detect the edge information of ARRA grapes and filter out most of the non-berry edge information such as stains and obtain relatively clear slender grape berries edges. But for ARRA grapes, some areas with larger stains or darker shades cannot be eliminated by filtering algorithms and small connected area elimination algorithms. The edge contours of these areas will be classified as effective contours. Secondly, the overlapping part of the berry can accurately estimate the position of its concave point through the curvature algorithm, and then remove the singular pixels that are not smooth in the contour segment. However, because the edge of the berry is not in a circular arc shape and the local maximum curvature of the partially intersecting area is not obvious, it cannot Accurately detect its concave point. Finally, the local clustering search strategy and the rotation direction judgment condition are to determine the berry group of the contour segment based on the arc information of the round berry. The edge information of the slender grape berry presents an irregular shape. The algorithm in this paper cannot accurately combine the contour segments for the same berry.

(3)The authors mention that there might be issues with "squeezed" grapes, but a section about the limitations, and ideas on how to possibly overcome these would be recommended to outline this further. 
 Responds: This article adds a description of the problem of "squeezing" grapes and stains in section 5.2. we have removed most of the stains through filtering algorithms and small connected areas, and we have also used edge curvature thresholds to make the contour segments relatively smooth to reduce the impact of compression deformation.

(4)line 78: remove "at home and abroad".
 Responds: This part has been changed to "Researchers have done a lot of research on this"

(5)line 91: "grapes berries" -> "grape berries"
 Responds: This part has been changed to "grape berries".

(6)line 100: new paragraph after "detected.".
 Responds: The part after "detected." has been divided into a new paragraph.

(7)line 110: incomplete sentences. Suggestion: "It solves the problem ..."
 Responds: This part has been changed to "It solves the problem ...".

(8)line 118: consider using "estimated" instead of "measured"
 Responds: This part has been changed to "estimated".

(9)line 122: --> "weather conditions" 
 Responds: "conditions" has been added after "weather"

(10)line 124: "as shown in ..." -> "Examples are shown in ..." 
 Responds: All "as shown in ..." in this article are modified to "Examples are shown in ..."

(11)line 133: add "The image of " (sentence is incomplete) 
Responds: This part has been changed to "The image of grape strings of various types" and "To reduce the interference effects such as noise on the grape berries, this paper uses bilateral filtering to remove the interference in the image while retaining more edge information in the image".

(12)Figure 4: a, b, c are not visible in the figure.
Responds: Already added "(a),(b),(c)" in Figure 4 of this article.

(13)line 144: In this section, we describe ... (add: we)
Responds: This part has been changed to "we".

(14)line 160: something is wrong with this sentence: "influence point subtracts ..."
Responds: This part has been changed to "If it is a negative influence point minus 1"

(15)line 218: "to the test": I did not understand this.
Responds: According to our many experimental tests, the best threshold is set to 0.015.

(16)line 370: "mention" -> "mentioned"
Responds: This part has been changed to "mentioned".

(17)Figure 10: some of the symbols in the caption are not visible in the figure.
Responds: Due to many pictures, some symbols are in a smaller area in the picture.

(18)Caption of Figure 11: R_1
Responds: Figure 11. Effect of different R1 on grape pulp detection. R1 is the fuzzy search range near the marked centroid point.
The analysis shows that different R1 has different effects on the TPR of grape berries.